# Native proline-rich motifs exploit sequence context to target actin-remodeling Ena/VASP protein ENAH

Theresa Hwang[1], Sara S Parker[2], Samantha M Hill[2], Robert A Grant[1], Meucci W Ilunga[1], Venkatesh Sivaraman[1], Ghassan Mouneimne[2], Amy E Keating[1,3]*

[1]Department of Biology, Massachusetts Institute of Technology, Cambridge, United States; [2]Department of Cellular & Molecular Medicine, University of Arizona, Tucson, United States; [3]Department of Biological Engineering and Koch Institute for Integrative Cancer Research, Massachusetts Institute of Technology, Cambridge, United States

*For correspondence:
keating@mit.edu

**Abstract** The human proteome is replete with short linear motifs (SLiMs) of four to six residues that are critical for protein-protein interactions, yet the importance of the sequence surrounding such motifs is underexplored. We devised a proteomic screen to examine the influence of SLiM sequence context on protein-protein interactions. Focusing on the EVH1 domain of human ENAH, an actin regulator that is highly expressed in invasive cancers, we screened 36-residue proteome-derived peptides and discovered new interaction partners of ENAH and diverse mechanisms by which context influences binding. A pocket on the ENAH EVH1 domain that has diverged from other Ena/VASP paralogs recognizes extended SLiMs and favors motif-flanking proline residues. Many high-affinity ENAH binders that contain two proline-rich SLiMs use a noncanonical site on the EVH1 domain for binding and display a thermodynamic signature consistent with the two-motif chain engaging a single domain. We also found that photoreceptor cilium actin regulator (PCARE) uses an extended 23-residue region to obtain a higher affinity than any known ENAH EVH1-binding motif. Our screen provides a way to uncover the effects of proteomic context on motif-mediated binding, revealing diverse mechanisms of control over EVH1 interactions and establishing that SLiMs can't be fully understood outside of their native context.

## Editor's evaluation

The manuscript uses a new screen called MassTitr to display long (36-mer) peptides derived from human proteome to screen for peptides that can bind the EVH1 domain of ENAH protein. About 100 peptides were identified and further analysis identified sequence features that contribute to the binding of EVH1 domain, including an additional proline after the FP4 motif and double FP4 motif. This paper will be of broad interest in the field of proteomics and to scientists interested in how biological interactions achieve specificity.

## Introduction

Interactions between modular interaction domains and short linear motifs (SLiMs) direct a broad range of intracellular functions, from protein trafficking to substrate targeting for post-translational modifications. To faithfully propagate signals, SLiMs must recognize the correct interaction partners within the cellular environment. But how interaction specificity is achieved is enigmatic. SLiMs, which occur

as 3–10 consecutive amino acids in intrinsically disordered regions of proteins, are degenerate and have low complexity, meaning they are defined by just a few key residues or motif features. Crystal structures of SH3, WW, and PDZ domains bound to SLiMs typically reveal three to six residues docked into a shallow groove (*Lim et al., 1994*; *Macias et al., 1996*; *Schultz et al., 1998*). The expansion of modular interaction domain families in metazoan proteomes has led to hundreds of domains that share overlapping SLiM-binding specificity profiles yet carry out distinct functions in the cell (*Bhattacharyya et al., 2006*). How high-fidelity interactions are maintained between low complexity SLiMs and cognate recognition domains remains poorly understood for many pathways.

Most SLiM research has centered around defining the 'core SLiM', or the minimal set of amino acids sufficient to bind to a given domain. High-throughput approaches, such as phage display using libraries of 7–16-residue peptides (*Ivarsson et al., 2014*; *Teyra et al., 2017*; *Tonikian et al., 2008*; *Davey et al., 2017*), have been instrumental for advancing our understanding. But these assays do not probe how the sequences surrounding core SLiMs affect their interactions, and there is increasing evidence that the surrounding sequence critically influences SLiM interaction affinity and specificity (*Palopoli et al., 2018*; *Prestel et al., 2019*; *Stein and Aloy, 2008*). For example, an alpha-helical extension C-terminal to a SLiM in ankyrin-G confers high-affinity and selective interactions with the GABARAP subfamily of Atg8 proteins by making contacts with the GABARAP interface (*Li et al., 2018b*). The presence of aromatic residues directly flanking a SLiM in Drebrin prevents its interaction with Homer, demonstrating that SLiM sequence context can also disfavor protein-protein interactions (*Li et al., 2019*).

The actin interactome contains many proline-rich SLiMs and many proline-binding modules such as SH3, WW, and EVH1 domains that participate in regulating actin dynamics (*Holt and Koffer, 2001*). Although the extent to which these domains cross-react or bind selectively in the cell is unknown, sequence elements surrounding linear, proline-rich motifs could play an essential role in directing specific interactions. Therefore, we sought to uncover the impact of sequence context on SLiM-mediated interactions with the EVH1 domain of the actin-regulating Ena/VASP protein ENAH.

Ena/VASP proteins form a family of cytoskeletal remodeling factors that are recruited to different regions of the cell by binding proline-rich SLiMs via their N-terminal EVH1 domains and promoting actin polymerization via their C-terminal EVH2 domains. The family is implicated in many cellular functions such as axon guidance and cell adhesion (*McConnell et al., 2016*; *Scott et al., 2006*). The Ena/VASP EVH1 domain recognizes the SLiM [FWYL]PX$\Phi$P, where X is any amino acid and $\Phi$ is any hydrophobic residue (*Ball et al., 2000*). This motif, referred to in this paper as the *FP4 motif*, because FPPPP is a common example, adopts a polyproline type II (PPII) helix structure and binds weakly to the EVH1 domain (*Prehoda et al., 1999*). Searching for this core FP4 motif in the human proteome yields 4,994 instances. This number of potential interaction partners is very large, and although spatial, structural, and temporal context impose additional determinants for cellular interaction (*Bugge et al., 2020*), the abundant motif matches raise the question of whether sequence elements beyond the FP4 SLiM affect molecular recognition.

We used a new screening approach to uncover examples of how the sequence context surrounding the core Ena/VASP FP4 SLiM affects binding specificity in the proteome. Our unbiased screening method, MassTitr, identified 36-residue human proteome-derived peptides that bind to the ENAH EVH1 domain with a range of affinities. To our knowledge, this is the first use of a high-throughput screening method to systematically discover and characterize both local and distal sequence elements that impact SLiMs. By analyzing features of high-affinity binders, we identified distinct ways in which sequence elements surrounding proteomic FP4 SLiMs impact binding affinity and specificity for ENAH. Our work provides insight into how selective interactions are maintained in proline-rich motif-mediated signaling networks and highlights the importance of considering sequence context when investigating SLiM-mediated interactions. Our pipeline serves as a blueprint to map and predict how the sequence context surrounding SLiMs impacts protein-protein interactions on a proteome-wide scale.

## Results

### MassTitr identifies ENAH EVH1 domain-binding peptides from the human proteome

To identify ENAH EVH1 binders in the human proteome, we applied a screen called MassTitr. MassTitr is a SORT-SEQ method that is based on fluorescence-activated cell sorting (FACS) of a library of peptide-displaying bacteria and subsequent deconvolution of signals by deep sequencing. As shown in *Figure 1A and B*, peptide-displaying *Escherichia coli* cells are sorted into bins according to their binding signals across a range of protein concentrations, and the binding signal for each peptide at each protein concentration is extracted by deep sequencing each bin. Two advantages of this method over phage display are that MassTitr supports screening of long peptides and leads to the identification of binders with a broad range of affinities. MassTitr is similar in concept to yeast-surface-display based methods that have been applied to study the interactions of anti-fluorescein scFvs and SARS-CoV-2 receptor binding domain mutants (*Adams et al., 2016*; *Starr et al., 2020*).

Using MassTitr, we screened a library of 416,611 36-mer peptides with seven-residue overlaps (the T7-pep library) (*Larman et al., 2011*). This library spans the entire protein-coding space of the human genome, and we hypothesized that the long lengths of the encoded peptides would illuminate the impact of the sequence surrounding the FP4 motif in a biologically relevant sequence space. We first prescreened the library for binding to an ENAH EVH1 domain that was tetramerized by fusion to the endogenous ENAH coiled coil, as shown in *Figure 1A*, generating an input library enriched in binders. We then ran MassTitr on the prescreened library, using eight concentrations of ENAH EVH1 tetramer (*Figure 1B*). After sorting, sequencing, and filtering based on read counts, 108 unique high-confidence binders were identified and classified as either high-affinity or low-affinity as described in the methods (*Figure 1C*, *Supplementary file 1*). Of the 108 hits, 14 may have bound to ENAH EVH1 because a library synthesis error introduced a motif that is not present in the human protein; we did not analyze these sequences further.

We validated the binding of 16 MassTitr peptide hits to monomeric ENAH EVH1 domain by using biolayer interferometry (BLI) to determine dissociation constants that ranged from 0.18 µM to 63 µM (*Supplementary file 2*). Except for SHROOM3 and TENM1 peptides, binders classified as high-affinity by MassTitr bound to the ENAH EVH1 domain more tightly than peptides classified as low-affinity. Many newly identified peptides bound with affinities similar to or tighter than a well-studied control peptide from *Listeria monocytogenes* protein ActA, which bound with $K_D$ = 4.9 µM in our BLI assay (*Supplementary file 2*). Prior to this work, this single FP4-motif-containing sequence from ActA was the tightest known endogenously derived binder of Ena/VASP EVH1 domains (*Ball et al., 2000*). The highest affinity peptide that we discovered was from photoreceptor cilium actin regulator (PCARE) ($K_D$ = 0.18 µM for 36-residue peptide PCARE[813-848]; *Supplementary file 2*), which contains the FP4 motif LPPPP. Successive truncations of this peptide identified the 23-residue minimal region for high-affinity binding, which extends 14 residues beyond the FP4 motif (PCARE[826-848] $K_D$ = 0.32 µM, *Figure 1— figure supplement 1*).

Although the majority of MassTitr hits contained FP4 motifs, 40 out of the 108 high-confidence hits did not (*Figure 1C and D*, *Supplementary file 1*). Most of the non-canonical binders contained a CXC motif. Although disulfide bond formation between CXC-containing peptides and ENAH may contribute to signal in the cell-surface display assay, we confirmed that this motif, and not just the presence of one or more cysteine residues, was important for the binding of a peptide from OLIG3 to ENAH in the presence of 2 mM DTT (*Figure 1—figure supplement 2*). Also, a CXC-containing peptide from TRIM1 bound reversibly to the ENAH EVH1 domain at mid-micromolar concentrations (*Figure 1—figure supplement 2*). Peptides from KIAA1522 and TJAP1 bound to ENAH and lacked either an FP4 or a CXC motif (*Supplementary file 2*). Our results, therefore, add to increasing evidence that the ENAH EVH1 domain can bind sequences beyond the FP4 motif (*Boëda et al., 2007*; *Chen et al., 2014*; *Menon et al., 2015*).

### MassTitr peptides are associated with and expand the ENAH signaling network

To highlight putative biologically relevant interaction partners of ENAH, we applied a bioinformatic analysis to identify those motifs that are likely to be accessible and co-localized with ENAH. We

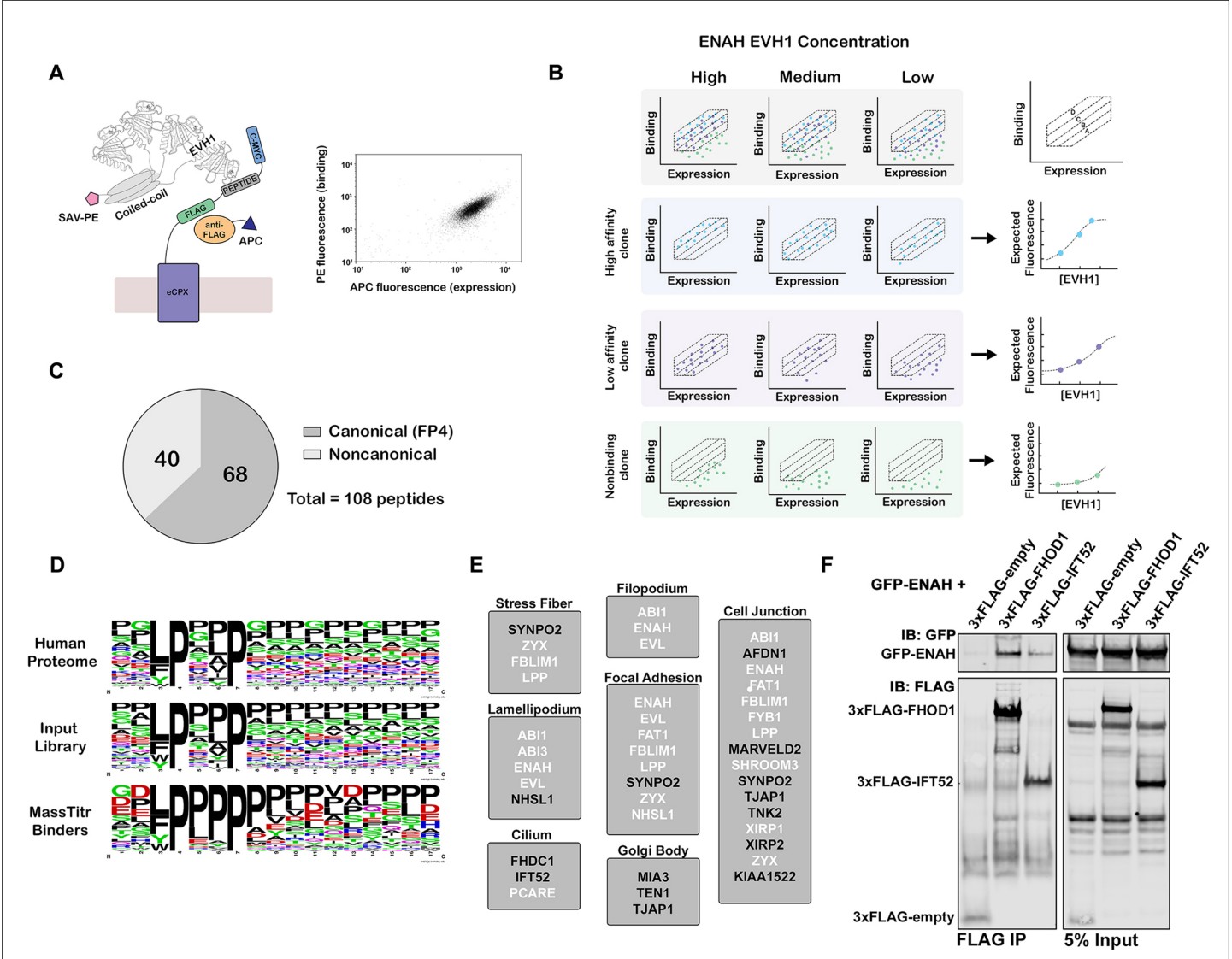

**Figure 1.** MassTitr screening identifies biologically relevant ENAH EVH1 ligands. (**A**) At left, bacterial surface display schematic. Library peptides flanked by a FLAG tag and a c-Myc tag were expressed as fusions to the C-terminus of eCPX on the surface of *E. coli*. Cells were labeled with anti-FLAG-APC to quantify expression and then incubated with tetrameric ENAH EVH1 domain, which was detected by streptavidin conjugated to phycoerythrin (SAV-PE). At right, a FACS plot for surface-displayed ActA peptide binding to ENAH EVH1 tetramer (10 μM monomer concentration). (**B**) MassTitr schematic. The top row represents a library of three clones (blue, purple, and green) sorted into four gates at three concentrations of ENAH. The rows highlighted in blue, purple, and green illustrate reconstructions of the concentration-dependent binding of each clone based on deep sequencing. The experiment in this paper sorted a pre-enriched library of clones into four gates at eight concentrations. (**C**) Distribution of MassTitr hits after filtering; 68 peptides contained a canonical FP4 motif matching the regular expression [FWYL]PX[FWYLIAVP]P. (**D**) Frequency plot made from sequences that match the FP4 motif in the human proteome, the input library, and the MassTitr binders using Weblogo (*Crooks et al., 2004*). (**E**) Subcellular locations where at least two MassTitr hits that are predicted to be disordered and localized in the cytoplasm are annotated to reside. White text denotes previously reported Ena/VASP interactions. (**F**) IP and western blot showing interaction of ENAH with MassTitr hits FHOD1 and IFT52 in cells. GFP-tagged ENAH and FLAG-tagged candidate interactors were overexpressed in cells and resulting lysate was precipitated with anti-FLAG antibody and then blotted with anti-FLAG and anti-GFP.

The online version of this article includes the following figure supplement(s) for figure 1:

**Figure supplement 1.** PCARE truncations.

**Figure supplement 2.** Peptides containing CXC motifs bind to ENAH EVH1 domain.

**Figure supplement 3.** Full-length ENAH interacts with TJAP1 in cells.

**Figure supplement 4.** Gates used for MassTitr FACS sorting.

filtered our high-confidence hits by disorder propensity (IUPred2A > 0.4) (*Mészáros et al., 2018*) and cytoplasmic subcellular localization (*Binns et al., 2009*; *Thul et al., 2017*). This resulted in 34 peptides that mapped to 33 unique proteins, of which 13 are derived from interaction partners previously known to interact or co-localize with an Ena/VASP protein (*Supplementary file 1*). The Ena/VASP binding sites of 10 of these hits have been previously mapped. Therefore, MassTitr provided new information about the EVH1-binding sites of 23 novel or previously known interaction partners of the Ena/VASP family. Filtered hits were highly enriched in GO biological process terms including actin filament organization (FDR < 10⁻⁶) and positive regulation of cytoskeleton (FDR < 0.05) (*Mi et al., 2019*), which align with documented cellular functions of ENAH. Notably, we also identified proteins localized to the Golgi body and cilia, where Ena/VASP function is not well characterized (*Figure 1E*; *Kannan et al., 2014*; *Tang et al., 2016*).

We tested whether putative new ENAH interaction partners bound to full-length proteins in mammalian cells. We overexpressed GFP-tagged full-length ENAH with FLAG-tagged FHOD1, IFT52, or TJAP1 and used an anti-FLAG antibody to precipitate complexes from the cell lysate. Probing with anti-FLAG and anti-GFP antibodies showed robust immunoprecipitation (IP) of GFP-ENAH relative to cells expressing GFP-ENAH and a FLAG-tag-only negative control protein (*Figure 1F*, *Figure 1— figure supplement 3*).

## A proline-rich C-terminal flank binds to a novel site on the EVH1 domain to enhance affinity in ENAH interaction Partners

We used MassTitr data to identify FP4 SLiM-flanking elements that enhance binding to the ENAH EVH1 domain. A sequence logo made of the high-confidence MassTitr hits shows enrichment of prolines C-terminal to the FP4 motif, and a binomial test confirms that peptides containing FP4 motifs followed by three consecutive prolines are enriched our hit list (p < 10⁻¹¹; *Figure 1D*). A peptide from ENAH interactor ABI1 (*Chen et al., 2014*; *Tani et al., 2003*) was among the highest affinity ligands that we validated by BLI, with $K_D$ = 2.6 µM (*Supplementary file 2*). ABI1 contains an FP4 motif followed by four prolines. Mutating FPPPPPPPP (FP₈) to FPPPPSSSS in the context of the ABI1 36-mer reduced affinity by approximately fourfold (p < 0.05; *Supplementary file 3*). Although this confirms that the C-terminal prolines enhance affinity, peptide FPPPPPPPP alone binds to the ENAH EVH1 domain with $K_D$ = 28 µM, indicating that additional interactions contribute to the high affinity of the ABI1 36-mer. Previous studies have shown that acidic residues N- and C-terminal to the FP4 motif can enhance affinity (*Niebuhr et al., 1997*) and we hypothesized that positively charged patches on ENAH could bind acidic residues that flank the FP₈ segment in ABI1 (*Figure 2—figure supplement 1*). Indeed, truncating the N-terminal or C-terminal acidic flanks of the 36-residue ABI1 peptide further decreased affinity (*Supplementary file 3*).

We solved a crystal structure of the ENAH EVH1-ABI1 peptide complex at 1.88 Å resolution. Only the FP₈ region was fully resolved in the structure under the crystallization conditions, which included high salt and low pH (*Figure 2A*). The peptide folds into a PPII helix with prolines 1, 4, and 7 (⁰F**P**PP**P**PP**P**P⁸) contacting the EVH1 surface (*Figure 2A and B*). The FP4 portion of the peptide binds the canonical FP4 groove, as observed in other structures (*Prehoda et al., 1999*; *Fedorov et al., 1999*), whereas the 7th proline docks into a previously uncharacterized site on ENAH composed of Ala12, Gly92, Phe32, and the aliphatic part of the side chain of Asn90 (*Figure 2C*). We note that residue 12 has diverged in EVL and VASP EVH1 domains (*Figure 2C*), which may contribute the weaker binding of ABI1 to those paralogs (*Supplementary file 4*), although we have not isolated the affinity difference to this specific change. Notably, a similar binding site at the analogous location is used by the Homer EVH1 domain to bind the phenylalanine of PPXXF motifs (*Beneken et al., 2000*). However, this site is relatively shallow in ENAH, and modeling large aromatic acids at this position on the ABI1 peptide using Pymol leads to severe steric clashes. Homer contains a smaller Gly89 at the site of Asn90 in ENAH and can accommodate the bulky Phe of the PPXXF motif (*Figure 2C*).

## Distal sequence elements enhance ENAH EVH1 binding through bivalent interactions

Another enriched feature of MassTitr-identified binders, relative to the pre-screened input library, is the presence of multiple FP4 motifs (binomial test, p < 10⁻²²). Multi-motif hits highlighted preferred spacings of approximately five or 15 residues between FP4 motifs (*Figure 3A*). Multiple motifs were

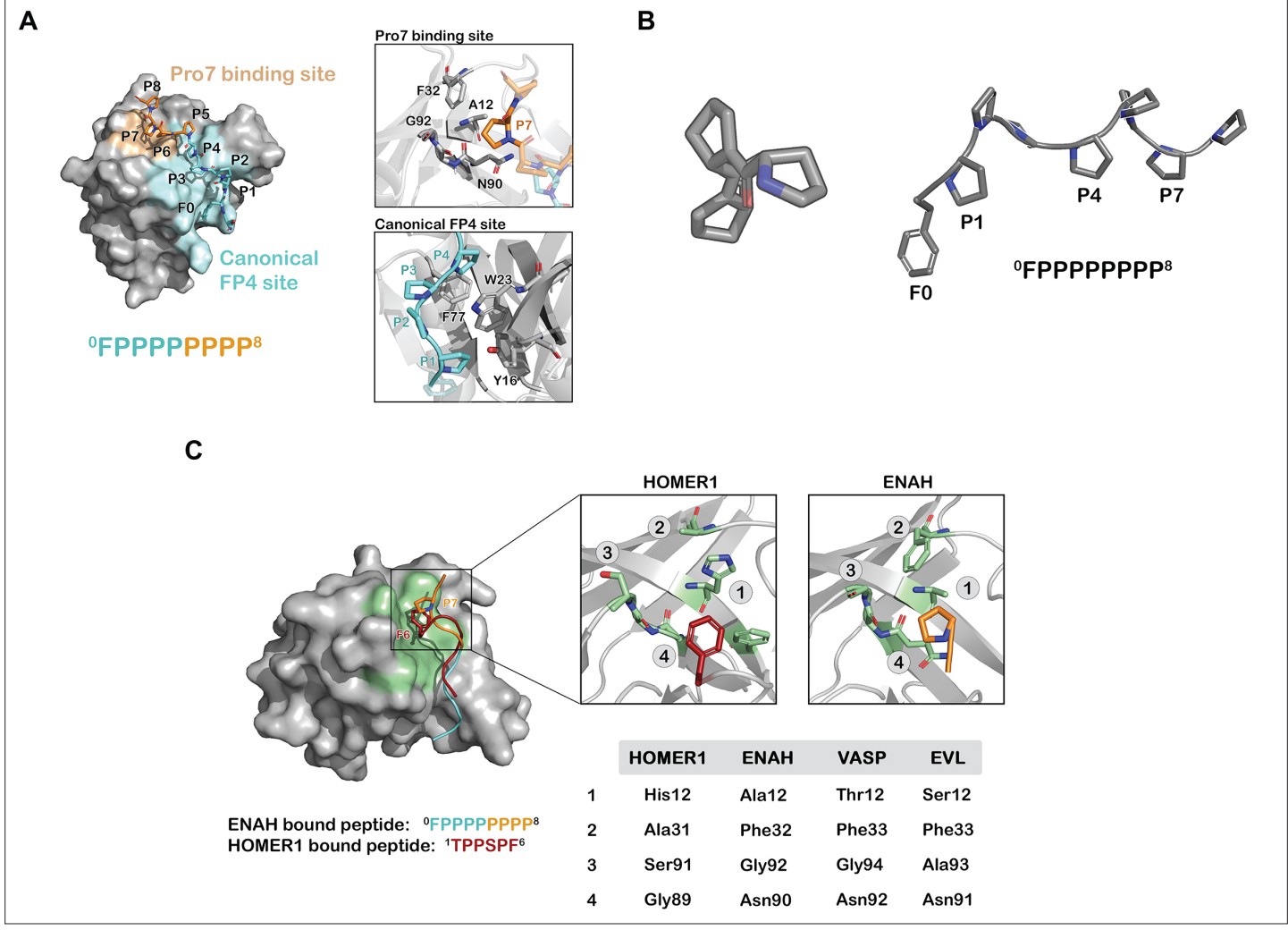

**Figure 2.** Prolines C-terminal to FP4 can engage a novel ENAH binding site. (**A**) Surface representation of the ENAH EVH1 domain bound to FP$_8$. The core FP4 motif is light blue, the P$_4$ flank is orange; insets show details of the interactions. (**B**) Axial view of a polyproline type II helix highlighting three-fold symmetry (left); a side view shows P1, P4, and P7 facing the same side (right). (**C**) At left, surface representation of the HOMER1 EVH1 domain bound to TPPSPF (PDB 1DDV, peptide in red) aligned to the ENAH EVH1 domain bound to peptide FP$_8$ (peptide in light blue/orange). The region corresponding to the Pro7 binding pocket in HOMER1 is colored in green. Inset: magnified views of the Pro7 binding pocket in ENAH and the analogous pocket in HOMER1. The table compares residues in this pocket for HOMER1, ENAH, VASP, and EVL.

The online version of this article includes the following figure supplement(s) for figure 2:

**Figure supplement 1.** Additional analyses of the ENAH EVH1-ABI1 structure.

also enriched in MassTitr high-affinity hits relative to all hits (p < 0.02), supporting the hypothesis that multiple FP4 motifs enhance affinity. We confirmed this experimentally by showing that binding was changed significantly (p < 0.05), with a 2.5- to sixfold reduction in affinity, when 36-mer sequences from LPP and NHSL1, which contain two FP4 motifs, were truncated to leave only one motif (*Table 1*, *Figure 3B*).

Zyxin, which contains four clustered FP4 motifs, has been shown to bind to the VASP EVH1 domain by contacting both the canonical FP4 site and a noncanonical site on the opposite side of the EVH1 domain (*Acevedo et al., 2017*). Interestingly, a crystal structure of the ENAH EVH1 domain bound to a single-FP4-motif peptide at the canonical site also contains a second peptide bound to the region corresponding to the noncanonical binding site in VASP (PDB 5NC7, *Barone et al., 2020*). To test whether multi-FP4-motif peptides engage this noncanonical site, we designed ENAH EVH1 R47A. In the ActA peptide-bound structure, ENAH Arg47 forms a bidentate hydrogen bond with a carbonyl on the peptide PPII helix backbone in the back-side site. The analogous VASP Arg48 exhibits significant

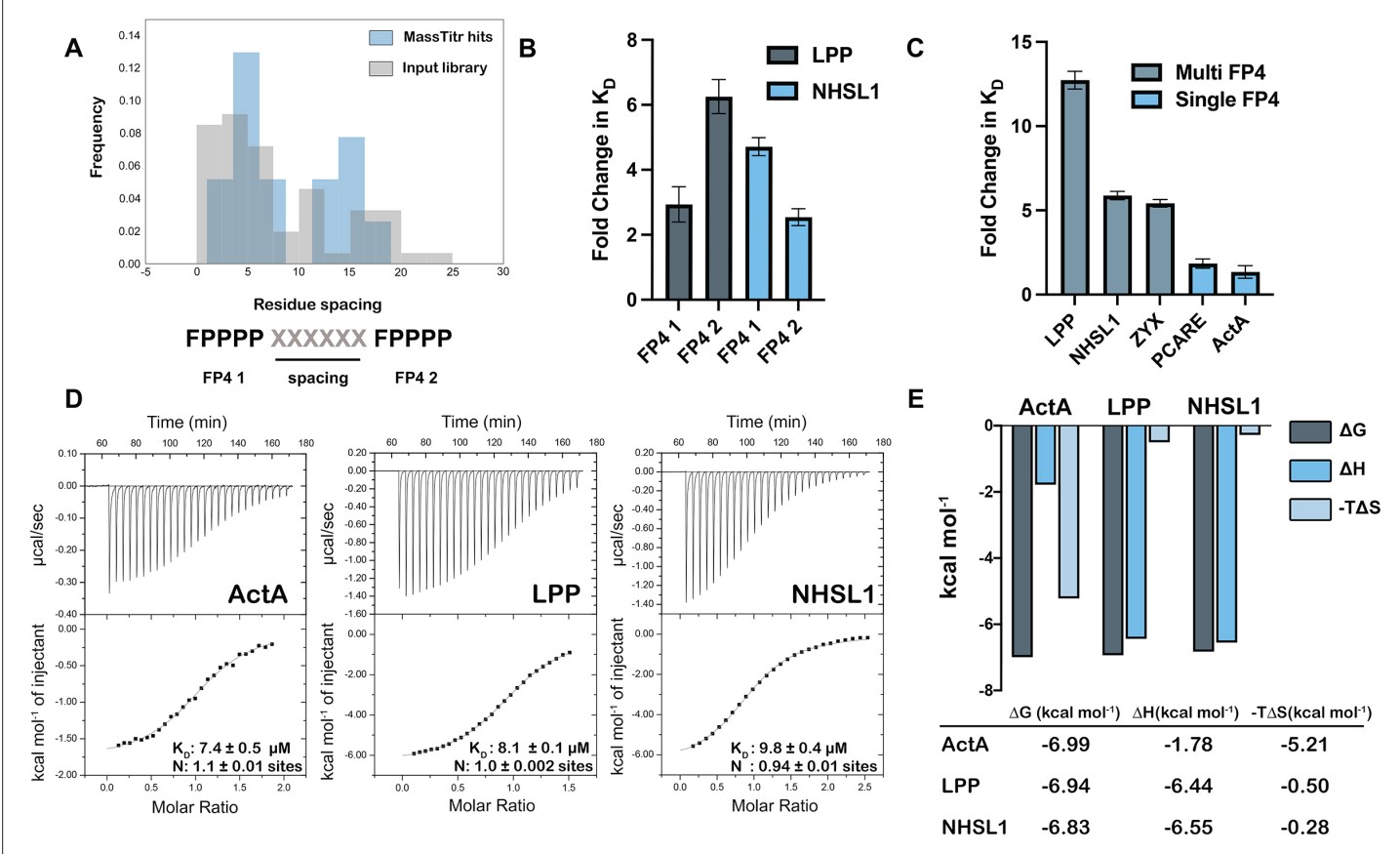

**Figure 3.** Multiple FP4 motifs enhance peptide binding affinity. (**A**) Spacing of FP4 motifs in the input library and in high-confidence hits. (**B**) Fold change increase in $K_D$ for truncated single-motif peptide variants relative to higher affinity 36-mer dual-motif library peptides for LPP and NHSL1; see *Table 1* for sequences. (**C**) Fold change increase in $K_D$ for 36-mer peptides binding to ENAH EVH1 R47A relative to tighter binding ENAH EVH1 WT. (**D**) ITC binding curves for 36-residue peptides from ActA, LPP, and NHSL1. (**E**) The entropic and enthalpic contributions to binding determined using data in panel D. Fold-change errors in (**B**) and (**C**) were calculated by propagating the error from two affinity measurements. Sequences for peptides referenced in this figure are given in *Table 1* and *Supplementary file 6*.

The online version of this article includes the following source data and figure supplement(s) for figure 3:

**Source data 1.** Raw data for *Figure 3B and C*.

**Figure supplement 1.** Modeling bivalent binding.

NMR HSQC chemical shifts upon titration with a multi-FP4 motif zyxin peptide (*Acevedo et al., 2017*). Thus, we predicted that the R47A mutation would disrupt back-site binding. Indeed, while the affinities of single-FP4-motif peptides from ActA and PCARE were minimally affected by this mutation, peptides from zyxin, LPP, and NHSL1 that contain two FP4 motifs showed a 5–15-fold reduction in affinity for ENAH upon mutating Arg47 to Ala (p < 0.01) (*Table 1*, *Figure 3C*, *Supplementary file 5*).

Next, we investigated the stoichiometry and thermodynamics of multi-motif peptide binding to a monomeric ENAH EVH1 domain. Using isothermal titration calorimetry (ITC), we confirmed that dual-FP4 motif peptides from LPP and NHSL1 and the single-FP4 motif peptide from ActA fit well to a 1:1 binding model (*Figure 3D*). Interestingly, the ITC analysis showed that binding of the ActA-derived single-FP4 motif peptide was driven by favorable entropy, whereas binding of the NHSL1 and LPP dual-motif peptides was enthalpically driven. ActA, LPP, and NHSL1 peptides have similar binding free energies, but the entropic contribution to the dual-motif interactions is ~10 fold less favorable (*Figure 3E*). These data are consistent with a model in which long, disordered dual-motif peptides pay an entropic penalty to wrap around the EVH1 domain and engage two sites but gain enthalpic binding energy from additional interactions. Duplication of the linkers between the two motifs in LPP and NHSL1 led to a very modest, ~ twofold reduction in affinity for the ENAH EVH1 domain compared

**Table 1.** Affinities of dual FP4 motif peptides and their variants for ENAH EVH1 WT or ENAH EVH1 R47A obtained using biolayer interferometry.

| Name[b‡] | Sequence | WT KD (µM) | R47A KD (µM) |
|---|---|---|---|
| NHSL1[*] | ADRSPF**LPPPP**PVTDCSQGSPLPHSPV**FPPPP**PEAL | 9.7 ± 2.5 | 51.5 ± 10.0 |
| NHSL1 FP4 1[*] | ADRSPF**LPPPP**PVTDCSQGSPLPHSPV | 45.9 ± 5.5 | 93.0 ± 16.0 |
| NHSL1 FP4 2[*] | PVTDCSQGSPLPHSPV**FPPPP**PEAL | 24.9 ± 1.2 | 53.0 ± 4.1 |
| NHSL1 Duplicated[*] | ADRSPF**LPPPP**PVTDCSQGSPLPHSPVPVTDCSQGSPLPHSPV**FPPPP**PEAL | 18.6 ± 0.2 | 65.0 ± 7.0 |
| LPP[*] | KQPGGEGDF**LPPPP**PPLDDSSALPSISGN**FPPPP**PL | 4.7 ± 2.4 | 60.1 ± 6.7 |
| LPP FP4 1[*] | KQPGGEGDF**LPPPP**PPLDDSSALPSISGN | 13.9 ± 2.5 | 61.5 ± 0.6 |
| LPP FP4 2† | PPLDDSSALPSISGN**FPPPP**PL | 29.6 ± 2.3 | 67.2 ± 14.9 |
| LPP Duplicated[*] | KQPGGEGDF**LPPPP**PPLDDSSALPSISGNDDSSALPSISGN**FPPPP**PL | 7.9 ± 2.2 | 53.3 ± 2.5 |

[*]Difference between WT and R47A KD value is significant with p < 0.01.

†Difference between WT and R47A KD value is significant with p < 0.05.

‡Errors are standard deviations over three replicates.

The online version of this article includes the following source data for table 1:

**Source data 1.** Raw data for *Table 1*.

to the WT LPP and NHSL1 peptides (p < 0.05 for NHSL1 duplicated, n.s. for LPP duplicated; *Table 1*). This is consistent with a less favorable conformational entropy of binding when the motifs are separated by a greater linker (*Table 1*). However, our data do not establish whether or not the two motifs bind the same way when the linker is duplicated. We observed that the interaction of truncated dual-motif peptides, which contain only a single motif plus the surrounding linker sequence, is weakened by mutation R47A (FP41 and FP42 peptides in *Table 1*). This suggests that the linker residues themselves may be able to make favorable interactions with the back-side site.

Finally, we examined the minimal motif-spacing requirements for bivalent binding. We used Rosetta to build a peptide chain to connect single FP4-motif peptides bound to the canonical and noncanonical sites of the ENAH EVH1 domain. There are two orientations of the chain that preserve the directionalities of the bound FP4 peptides observed in structure 5NC7. Ten residues were required to span the two motifs in orientation 1, whereas nine residues were sufficient in orientation 2 (*Figure 3—figure supplement 1*). This indicates that the ~15-residue spacing that was enriched in our hits is more than enough to span the two binding sites and implies that the chain is not taught between these two sites.

## Discussion

In recent years, phage display screening of peptides derived from the human proteome has been used to define SLiM specificity profiles and predict novel interaction partners (*Davey et al., 2017*; *Ueki et al., 2019*; *Jespersen et al., 2019*; *Wigington et al., 2020*); these studies have primarily focused on defining the "core SLiM". In this work, we used MassTitr to screen more than 400,000 36-residue segments of the human proteome against the cytoskeleton regulator ENAH. Analysis of the hits readily identified both local and distal sequence features up to 15 residues away from the core FP4 SLiM that are important for binding. Our study highlights ways in which low-information SLiMs exploit sequence context to selectively recognize modular interaction domains within the proteome, especially in the context of proline-rich signaling networks where over 300 SH3 domains, 80 WW domains, and 20 EVH1 domains coexist to drive signal transduction in humans (*Zarrinpar et al., 2003*).

We found multiple ways that sequence flanking the FP4 motif can modulate binding to ENAH. We first demonstrated that prolines C-terminal to FP4 motifs can enhance binding by contacting a previously uncharacterized hydrophobic patch on ENAH. Both secondary structure and sequence are key to this binding mode, which positions the 7th proline of a $^0$FPPPPPPP**P**P$^8$ peptide to contact ENAH in a shallow groove that we refer to as the Pro7 binding pocket. The relatively flat surface of the EVH1 domain in this region limits the binding energy available from favorable contacts, but PPII helix preorganization presumably minimizes the entropic cost of binding. We anticipate that this binding mode is widely exploited by cellular interaction partners of ENAH. Multiple previously annotated Ena/VASP

interactors, including proteins identified in our screen such as FBLIM1, ZYX, and LPP (*Zhang et al., 2006*; *Drees et al., 2000*; *Petit et al., 2000*) contain FP4 motifs followed by trailing prolines, with either a leucine or proline in the 7th position ([0]FPPPPPP**P**P[8]).

As shown in *Figure 2C*, the Homer EVH1 domain uses a site structurally analogous to the ENAH Pro7 binding pocket to accommodate Phe in the SLiM PPXXF (*Beneken et al., 2000*). Thus, part of the core binding site in the Homer EVH1 domain is used by the ENAH EVH1 domain as a secondary affinity-enhancing site. Also, interestingly, the Ena/VASP family members VASP and EVL have polar Thr or Ser residues in place of Ala 12 in this pocket. Both of these proteins bind ~5 fold less tightly than ENAH does to ABI1 (*Supplementary file 4*). The unique hydrophobic pocket on ENAH EVH1 provides a striking example of how a peripheral site that has diverged among homologous domains can engage motif-flanking sequence, which may provide a mechanism for increasing molecular recognition specificity.

Many known Ena/VASP partners contain multiple FP4 motifs (*Hansen and Mullins, 2015*) and dual-motif peptides were prevalent among our high-affinity hits, consistent with the multiple motifs enhancing binding. Analysis of our multi-FP4 MassTitr hits showed preferential spacing of ~5 or ~15 residues between FP4 motifs in a single chain. For peptides with a motif spacing of ~15 residues, our data and previous work support a model of bivalent binding, where multi-FP4 motif peptides can engage two sites on a single EVH1 domain (*Acevedo et al., 2017*; *Barone et al., 2020*). A 1:1 stoichiometry is supported by ITC for dual-motif peptides LPP and NHSL1 binding to a monomeric EVH1 domain, in an interaction that is weakened by disruption of the noncanonical site by mutation R47A. Doubling the linker length between motifs weakened binding, slightly, consistent with alteration of the effective concentration of a bivalent interaction (*Table 1*).

We speculate that diverse sequences, particularly those with the propensity to adopt a PPII helix conformation, could make favorable contacts with the back-side site when present at high effective concentration due to the binding of a primary FP4 motif at the canonical site. In support of this, we saw that the peptides from LPP and NHSL1 that lacked a second FP4 motif but contained at least one proline residue ~10 residues away from a single-FP4 motif bound at least twofold more tightly to wild-type ENAH EVH1 than to ENAH EVH1 R47A (*Table 1*). Our data suggest that either a second FP4 motif or linker residues can make favorable interactions with the noncanonical site.

FP4 motifs separated by five residues probably do not bind simultaneously to a single EVH1 domain, as structural modeling suggests that the minimum chain length required to span the two putative bindings sites is nine residues. In such cases, it may be that two EVH1 domains bind to two closely spaced motifs (see one possible model in *Figure 2—figure supplement 1B*). Another possibility is that clustered FP4 motifs separated by only a few residues bind using mechanisms such as allovalency, where the increased effective concentration of multiple FP4 motifs close together enhances affinity (*Levchenko, 2003*).

The critical noncanonical site residues for binding FP4 motifs, including ENAH Arg47 and VASP Tyr38 (*Acevedo et al., 2017*), are conserved across ENAH, VASP, and EVL, suggesting that bivalent binding is a general mechanism to increase molecular recognition specificity for the Ena/VASP family. However, there is also some evidence that this binding mode could provide paralog specificity, as the linker region connecting multiple FP4 motifs could contact regions on the EVH1 domain that differ across the Ena/VASP paralogs. In support of this, we found that a dual-FP4-motif peptide from LPP bound ~7 fold tighter to ENAH over EVL EVH1 domains (*Supplementary file 4* and p < 0.01).

Finally, we identified a peptide derived from PCARE that binds to ENAH with the highest known affinity of any SLiM ($K_D$ = 0.18 μM). Truncation experiments indicated that the 14-residues C-terminal of the LPPPP motif in PCARE are critical for its high affinity, hinting that extensive contacts between this region and the ENAH EVH1 domain could be responsible for the enhanced binding. Our subsequent work revealed the surprising structural basis for this affinity (*Hwang et al., 2021*).

Filtering MassTitr hits for interactions of most probable biological significance, based on localization and disorder, yielded peptides from 33 putative binding partners. 19 proteins from this list have not, to our knowledge, been reported to associate with Ena/VASP proteins and provide avenues for further investigation. Some of the binding partners that we discovered lack a match to the canonical FP4 motif. The segment from TJAP1 that gave a hit in our screen and was confirmed to bind to ENAH in IP experimentsdoes not contain any recognizable FP4 motif yet is proline-rich and binds to the ENAH EVH1 domain with a $K_D$ of 23 μM by BLI (*Supplementary file 2*). We also confirmed that a

region from KIAA1522 that lacks an FP4 motif but includes 10 proline residues in the 36-mer peptide binds to the ENAH EVH1 domain ($K_D$ = 14 µM; *Supplementary file 2*). KIAA1522 potentiates metastasis in esophageal carcinoma and breast cancer cells (*Xie et al., 2017*; *Li et al., 2018a*), potentially linking ENAH and KIAA1522 in tumor progression. Our results imply that while the sequence context surrounding FP4 motifs can significantly impact their affinity and specificity to the Ena/VASP family, noncanonical motifs also contribute to the Ena/VASP interactome.

We were able to verify interactions of ENAH with FHOD1, IFT52, and TJAP1 by co-IP assay in mammalian cells (*Figure 1F*, *Figure 1—figure supplement 3*). TJAP1 is primarily localized to the trans-Golgi complex and is thought to help maintain Golgi body structure (*Tamaki et al., 2012*). ENAH has been shown to regulate Golgi architecture in *Drosophila* photoreceptors and to play a role in maintaining Golgi structure in human cells via its interaction with GRASP65 (*Kannan et al., 2014*; *Tang et al., 2016*). However, the role of Ena/VASP proteins in regulating functions of the Golgi body is largely unexplored, positioning TJAP1 as a promising lead to further explore the role of Ena/VASP proteins in the Golgi body.

FHOD1 is one of several formin proteins (FHOD1, FHDC1, FMN2) that were identified as putative interactors in our screen. Like Ena/VASP proteins, formins also promote unbranched actin polymerization. There is evidence that the two families cooperate in regulating filopodial protrusions (*Barzik et al., 2014*), although the mechanistic basis behind this interaction is not well understood. Our hits are potential leads to further investigate the intersection between formins and Ena/VASP proteins in fine-tuning filopodial formation and dynamics.

IFT52 is part of the intraflagellar transport B complex (IFT-B) and is critical for the assembly of cilia and flagella. Mutations in the IFT-B complex are associated with several ciliopathies. IFT52 has been linked to short-rib thoracic dysplasia and retinal ciliopathies (*Chen et al., 2018*). To date, the role of Ena/VASP proteins in cilia has not been well characterized, although PCARE, a cilia-associated protein primarily found in the outer segment of photoreceptor cells, has been reported to associate with ENAH through tandem affinity purification mass spectrometry (*Corral-Serrano et al., 2020*). PCARE was also a hit from our MassTitr screen, pointing to a significant as-yet unexplored role for Ena/VASP proteins in cilia.

## Conclusion

For many protein domains beyond EVH1, degenerate SLiMs have been cataloged in the Eukaryotic Linear Motif (ELM) database to describe their interaction preferences (*Kumar et al., 2022*). The ELM listing implies that there is a relatively simple recognition code for many key domain interactions. However, the short sequences of most SLiMs are likely insufficient for biological specificity in many or most cases. Here we showed how defining the EVH1 binding motif as [FWYL]PXΦP is an oversimplification and how, by systematically examining the role of flanking sequences for just one EVH1 domain, we readily uncovered numerous examples in which the binding is modulated via additional extra-motif residues. Added to prior reports from investigations of individual interactions (*Stein and Aloy, 2008*; *Li et al., 2018a*; *Aitio et al., 2010*), our work definitively demonstrates the importance of sequence context on SLiM behavior by illustrating specific mechanisms, including an unusual conformational specificity mechanism that is documented in our companion paper (*Hwang et al., 2021*). MassTitr provides a versatile experimental platform for uncovering context effects on domain-peptide interactions and will surely lead to similar insights into the recognition strategies of other domains.

# Materials and methods

**Key resources table**

| Reagent type (species) or resource | Designation | Source or reference | Identifiers | Additional information |
|---|---|---|---|---|
| Strain, strain background (*Escherichia coli*) | DH5a | NEB | Cat# 2987 H | Chemically competent cells |
| strain, strain background (*Escherichia coli*) | BL21(DE3) | Novagen | Cat# 71,400 | Chemically competent cells |
| Peptide, recombinant protein | Streptavidin, R-Phycoerythrin Conjugate (SAPE) | Thermo Fisher | Cat# S866 | (1:100) |

*Continued on next page*

*Continued*

| Reagent type (species) or resource | Designation | Source or reference | Identifiers | Additional information |
|---|---|---|---|---|
| Antibody | SureLight Allophycocyanin-anti-FLAG antibody (mouse monoclonal) | Perkin Elmer | Cat# AD0059F | (1:100) |
| Antibody | Anti-FLAG (mouse monoclonal) | ProteinTech Group | Cat# 66008–3, RRID:AB_2749837 | (5 µg per mg of protein) |
| Antibody | Anti-FLAG (rabbit polyclonal) | ProteinTech Group | Cat# 66002–1, RRID:AB_11232216 | (1:1000) |
| Antibody | Anti-GFP (mouse monoclonal) | ProteinTech Group | Cat# 66002–1, RRID:AB_11182611 | (1:1000) |
| Antibody | Anti-Mouse IgG Alexa Fluor 680 (goat polyclonal) | ThermoFisher Scientific | Cat# A21057 | (1:20,000) |
| Antibody | Anti-Rabbit IgG Alexa Fluor 790 (goat polyclonal) | ThermoFisher Scientific | Cat# A11367 | (1:20,000) |
| Cell line (*Homo-sapiens*) | HEK293T | ATCC | | |

## Biolayer interferometry (BLI) and data analysis

BLI was carried out as described in *Hwang et al., 2021*. Briefly, biotinylated 6x-His-SUMO-peptide fusions were immobilized on streptavidin-coated tips and immersed into a concentration series of monomeric ENAH EVH1 domain diluted in BLI buffer (PBS pH 7.4, 1% BSA, 0.1% Tween-20, 1 mM DTT). Data were collected until the binding signal plateaued. Tips were then placed into 200 µL of BLI buffer, to allow dissociation, and data were collected until the signal plateaued. Quantification of the steady-state binding signal was performed to obtain $K_D$ values. Association curves were fit to a one-phase association model in Prism as given below:

$$Y = Y_o + \left( Plateau - Y_o \right) * \left( 1 - e^{-K*X} \right)$$

and equilibrium-bound signal values (Plateau) were plotted against ENAH concentration and fit to a single-site binding model:

$$Y = \frac{B_{max} * X}{K_D + X}$$

to obtain dissociation constants. The kinetics of association and dissociation were too fast to fit accurately. Errors are reported as the standard deviation of two to three replicates (see source data, which includes confidence intervals; we always collected three replicates for cases where we made quantitative comparisons). For the mean $K_D$ of replicates, 95% confidence intervals were calculated by assuming a t-distribution.

Fold-change errors given in *Figure 3B and C* were calculated through error propagation using the formula:

$$\sqrt{\left( \frac{SD_a}{\mu_a} \right)^2 + \left( \frac{SD_b}{\mu_b} \right)^2}$$

An unpaired, two-tailed t-test was used to calculate whether the difference between two $K_D$ values was statistically significant.

## Protein expression and purification

Sequences for proteins used in this study are given in *Supplementary file 6*. Human ENAH EVH1 domain, followed by a 6x-Gly linker and ENAH mouse coiled coil (for tetramerization), were cloned into a pDW363 biotinylation vector that includes a C-terminal biotin acceptor peptide (BAP) tag and a 6x-His tag. This ENAH tetramer construct was expressed in Rosetta2(DE3) (Novagen) cells in Terrific Broth (TB) with 100 µg/mL ampicillin, 25 µg/mL chloramphenicol, and 0.05 mM D-(+)-biotin (for in vivo biotinylation). Cells were grown at 37 °C with shaking to an optical density at 600 nm (O.D. 600) of 0.5–0.7 and then induced with 1 mM IPTG and grown at 37 °C for 5 hr. One L of cells were then spun down and resuspended in 25 mL of binding buffer (20 mM Tris pH 8.0, 500 mM NaCl, 5 mM imidazole), and frozen at –80 °C overnight. The next day, pellets were thawed and supplemented with 0.2 mM phenylmethylsulfonyl fluoride (PMSF) protease inhibitor. Cells were sonicated ten times

for 30 s followed by 30 s of rest on ice and then centrifuged. The clarified lysate was filtered through a 0.2 μm filter and applied to 2 mL of Ni-nitrilotriacetic (Ni-NTA) acid agarose resin (GoldBio) equilibrated in wash buffer (20 mM Tris pH 8.0, 500 mM NaCl, 20 mM imidazole). The resin was then washed three times with 8 mL wash buffer and eluted with 10 mL of elution buffer (20 mM Tris pH 8.0, 500 mM NaCl, 300 mM imidazole). The elution was run through a S75 26/60 size exclusion column equilibrated in gel filtration buffer (20 mM Tris pH 8.0, 150 mM NaCl, 1 mM DTT, 10% glycerol). Purity was verified by SDS-PAGE, and the fractions were pooled, concentrated, and flash-frozen at –80 °C.

SUMO-peptide fusions were cloned into a pDW363 vector that appends a BAP sequence and 6x-His tag to the N-terminus of the protein and transformed into Rosetta2(DE3) cells. For ITC experiments, these cells were expressed in TB supplemented with 100 μg/mL ampicillin, grown to an O.D. 600 of 0.5–0.7, and induced with 1 mM IPTG. Induced cultures were purified as described above for the pMCGS7 constructs, except for the TEV cleavage step. Instead, after elution with elution buffer, the sample was directly applied to the S75 26/60 column equilibrated in gel filtration buffer. Fractions were pooled, concentrated, and flash frozen at –80 °C.

Monomeric EVH1 domain and small-scale preparations of biotinylated SUMO peptides used for BLI were purified as described in *Hwang et al., 2021*.

## Bacterial cell surface display plasmids and T7-Pep library cloning

Control peptides for display were expressed at the C-terminus of eCPX in a vector designed by the Daughtery group (*Rice and Daugherty, 2008*). This construct was modified to include a FLAG tag at the N-terminus of the peptide and a c-Myc tag at the C-terminus. The T7-pep library plasmids (gift from Elledge lab, Harvard University and Brigham and Women's Hospital), were transformed into Pir1 cells (ThermoFisher), grown, and miniprepped (Qiagen) to isolate the library plasmid. Plasmids were then cut with EcoRI (NEB) and XhoI (NEB), and the inserts were gel purified, combined, and concentrated with a Zymo Clean and Concentrate column and eluted with 50 μL of sterile MilliQ Water. To clone the T7-pep library into the eCPX vector, we first grew up 200 mL of DH5a cells containing an empty eCPX vector at 37 °C overnight. This culture was miniprepped (Qiagen) and then digested with EcoI and XhoI at a ratio of 10 units of enzyme:1 μg of vector at 37 °C for 2 hr. The resulting digest was PCR purified and eluted with 40 μL water. This cut vector was then dephosphorylated with Antarctic phosphatase (NEB) at a ratio of 1 μL/1 μg of DNA at 37 °C for 2 hr, followed by 10 min at 65 °C for enzyme inactivation. T7-pep library insert was ligated into the cut eCPX vector using T4 ligase (NEB) at 14 °C overnight. Ligase was subsequently deactivated for 10 min at 70 °C, and the ligation reaction was concentrated with Zymo Clean and Concentrate columns (Zymo Research). Each column was eluted with 12.5 μL elution buffer (from kit). The resulting elutions were desalted on a 0.025 μm filter (Millipore) for 15–20 min and pooled on ice. Electrocompetent MC1061 cells and 10–20 μL DNA were then mixed and transferred to a cold 2 mm cuvette (BioRad). Each cuvette was pulsed at 2.5 kV, 50 μF, 100 ohms on an electroporator (BioRad), immediately rinsed out with 3 × 1 mL of warm SOC and transferred to a culture tube containing 7 mL warm SOC. Cells were incubated at 37 °C for 1 hr and then combined. Serial dilutions of the library were plated on LB/chloramphenicol plates to assess transformation efficiency, and the leftover cells were added to 500 mL of LB + 25 μg/mL chloramphenicol + 0.2% w/v sterile-filtered glucose. The library was grown at 37 °C until it reached an O.D. 600 of 2.0 and then frozen as glycerol stocks to use for FACS analysis and sorting.

## FACS sample preparation and analysis

The protocol for sample preparation for FACS analysis and MassTitr sorting was adapted from *Foight and Keating, 2016* and was as follows: 5 mL cell cultures of eCPX plasmid expressing either library or control peptide were grown overnight at 37 °C in LB + 25 μg/ml chloramphenicol + 0.2% w/v glucose. The next day, cells were inoculated into fresh TB + 25 μg/mL chloramphenicol and grown at 37 °C. Upon reaching an O.D. 600 of 0.5–0.6, cells were induced with 0.04% w/v arabinose for 1.5 hr at 37 °C. The O.D. 600 was then remeasured, and enough cells were pelleted for analysis (1 × $10^7$ cells per FACS analysis sample, 7 × $10^7$ cells for library sorting). Cells were resuspended to a concentration of 4 × $10^8$ cells/mL, washed in PBS + 0.1% BSA, and then incubated with anti-FLAG antibody conjugated to APC (αFLAG-APC; PerkinElmer) diluted 1:100 in PBS + 0.1% BSA at a ratio of 30 μL labeled antibody:$10^7$ cells. Tubes wrapped in foil were incubated at 4 °C for 15 min, then cells were washed with PBS + 0.1% BSA and pelleted. For each FACS analysis sample, 25 μL of 1 × $10^7$ cells in PBS

were mixed with 25 µL of a 2 x concentration of ENAH tetramer in PBS + 1% BSA + 4 mM DTT (final concentration of 2 mM DTT) and then incubated at 4 °C for 1 hr in foil. After incubation, 50 µL of the mixture was added per well to a 96-well Multi-Screen HTS GV sterile filtration plate (Millipore), buffer was removed by vacuum, and then cells were washed twice with 200 µL of PBS + 0.5% BSA. Each well containing $1 \times 10^7$ cells was then resuspended in 30 µL of streptavidin-PE (SAV-PE; ThermoFisher Scientific) diluted 1:100 in PBS + 0.1% BSA, and incubated for 15 minutes at 4 °C, washed with 200 µL of PBS + 0.1% BSA, and resuspended in 250 µL of PBS + 0.1% BSA for subsequent FACS analysis or sorting. For cell sorting, see supplementary methods.

## Pre-enrichment of T7-Pep library

The T7-pep library was prepared as described above. A total of $7 \times 10^7$ cells expressing the T7-pep library were incubated with a final concentration of 20 µM ENAH monomers assembled as tetramers (i.e. 20 µM monomer). Cells were sorted on a BDFACS Aria machine. Prior to sorting, a positive (ActA) control and a negative (empty) control were analyzed. A gate to collect cells expressing peptide binders was set that included 0.3% of the negative control and all cells with a greater binding signal, allowing us to enrich moderate-affinity binders. This process was repeated three times on the same day to collect a total of 300,000 cells. These cells were added to warm SOC + 25 µg/mL chloramphenicol and grown at 37 °C overnight. The next day, cells were frozen at –80 °C as glycerol stocks for MassTitr sorting.

## MassTitr sorting scheme

The pre-enriched T7-Pep Library was used, as described in the section *FACS sample preparation and analysis*, above. A total of $7 \times 10^7$ cells stained with anti-FLAG APC were incubated with increasing concentrations of biotinylated ENAH EVH1 tetramer (monomer concentrations: 30, 12, 4.8, 1.9, 0.77, 0.31, 0.12, 0.049 µM), and then incubated with streptavidin-PE. Labeled cells were sorted on a BDFACS Aria. Cells were collected in four gates that were drawn with boundaries roughly parallel to the binding vs. expression signal slope for a series of positive controls (ActA, SHIP2, Vinculin). The gates were drawn to evenly sample the binding signal range, in log space, for low-, medium-, and high-affinity clones across the FACS window, as shown for gates labeled A-D in *Figure 1—figure supplement 4*. A no-peptide negative control (empty) was run to assess the degree of nonspecific binding at the time of sorting. At each concentration of ENAH EVH1 domain, cells were sorted for approximately 20 min and the number of cells that were collected in each gate was recorded. Enough cells were collected per gate to oversample the library at least 10-fold. Cells were collected in LB + 0.2% w/v glucose and then sorted cells were transferred into 10 mL of LB + 0.2% w/v glucose, grown at 37 °C overnight, and plasmid DNA was isolated by miniprep the next day. Three replicate MassTitr experiments were performed.

## Illumina amplicon preparation

As described above, sorted pools were grown overnight at 37 °C in LB + 0.2% glucose and then miniprepped (Qiagen). Samples collected at a certain gate per concentration of ENAH tetramer were assigned a unique barcode/index combination. First, we PCR amplified the variable region of the library with a forward primer (Ngsfwd_1) and a corresponding reverse primer that contained one of 5 6-nucleotide (nt) index sequences for multiplexing (Ngsrev_1_i); the amplicon preparation scheme and all primers are listed in *Supplementary file 7*. Fourteen cycles of amplification were carried out with Phusion polymerase (NEB) using an annealing temperature of 66 °C. The resulting reaction was PCR purified with the Zymo Clean and Concentrate kit (Zymo Research) and eluted with 20 µL milliQ water. 200 ng of each reaction was cut with MmeI (NEB) at 37 °C for 1 hr and then the enzyme was heat inactivated at 65 °C for 20 min. To the 5' end of 15 µL of digested fragment we ligated a double stranded adapter with matching overhangs using T4 DNA ligase (NEB) at 25 °C for 30 mins, followed by heat inactivation at 65 °C for 10 min. This adapter contained one of 24 5-nt barcodes and the standard Illumina forward primer. The expected ~200 bp band was gel purified with the Zymoclean Gel DNA recovery kit (Zymo Research) and eluted in 17 µL of MilliQ water. Three µL of this reaction was PCR amplified for 10 cycles with primers Ngsfwd_2 and Ngsrev_2 in a 50 µL reaction at an annealing temperature of 66 °C. The 5' and 3' Illumina adapter sequences and the reverse priming sequence were included in this step. The final product was PCR purified and eluted in MilliQ water. The DNA

concentration of each pool was measured with the Qubit assay and samples were combined and run in one lane. In total, the multiplexed sample included 32 pools (corresponding to cells collected at each of eight concentrations in four gates) for each of three replicates, as well as the pre-enriched input library, for a total of 97 samples distinguished by their barcode/index combination. At each stage of the preparation process, the quality and homogeneity of the amplicon were assessed through Sanger sequencing (Genewiz) and Bioanalyzer. The multiplexed sample was submitted for sequencing on a NextSeq500 instrument using paired-end reads. *Supplementary file 7* gives an overview of the procedures used to (1) label DNA from each pool of cells with a barcode/index indicating the ENAH EVH1 tetramer concentration and gate number for the sample and (2) to prepare the library for sequencing; primer sequences are listed, as well as the barcode/index key used for each sample.

## MassTitr Data Processing

Sequences were demultiplexed and processed with an in-house script. . We used the sequencing data to determine the number of clonal cells (i.e., cells displaying the same peptide) that were found in each gate at each concentration. We calculated the clone read frequency in gate j at concentration k as $R_{ijk}/T_{jk}$, where $R_{ijk}$ is the number of raw reads for sequence i in gate j at concentration k, and $T_{jk}$ is the total sequencing reads for all clones obtained for gate j at concentration k, i.e., $T_{jk} = \sum_i R_{ijk}$ . The clone read frequency was multiped by the total number of cells collected in each gate (which was recorded during the sorting) to calculate the total number of cells displaying sequence i, $C_{ijk}$, in gate j at each ENAH concentration k. This can be used to compute an effective gate position using the following:

$$\sum_j C_{ijk} F_j / \sum_j C_{ijk} = F_{eff}^{i,k}$$

Where $F_j$ values are the mid y-axis fluorescence values for each gate, j = 1–4 (with values of 700, 1500, 2500, 4,000 respectively). $F_{eff}^{i,k}$ was calculated for each clone across all ENAH tetramer concentrations. If this experiment is performed in such a way such that every peptide-expressing cell is collected and sequenced, then the clones-per-gate information can be used to extract a concentration-dependent signal-vs.-concentration curve that can be fit to give an apparent dissociation constant, as done by Adams, et al. (*Adams et al., 2016*). In our experiment, we collected and sequenced fewer cells, focusing on cells that displayed above-background binding signal, and obtained information about concentration-dependent binding but not complete titration curves.

Our data processing was focused on identifying a subset of well-behaved clones, of varying affinities, that we judged to be good candidates for subsequent analysis of binding features. We first removed clones for which we obtained < 100 reads across all bins at all concentrations and also clones that did not have a total cell count ($C_{ijk}$ summed over all j) of at least 25 cells in at least four concentrations k. A sequence was assigned as an ENAH binder if (1) the clonal population of cells showed a concentration-dependent increase in binding signal in at least two of three replicate experiments, or (2) the displayed peptide contained an FP4 motif and showed concentration-dependent binding in at least one replicate. Among binders identified in this way, clones were further tagged as "high affinity" if they met additional requirements: (1) total cell count of greater than or equal to 80 cells for at least four concentrations, (2) greater than or equal to 10 cells found in binding gates at the lowest three concentrations (0.31 µM, 0.12 µM, 0.049 µM), (3) cells in more than one gate at the highest two concentrations (30 µM, 12 µM), and (4) clone found in at least two replicates. These filters are were chosen, based on benchmarking against validated binding clones from the screen, to extract a subset of well-behaved clones that we judged likely to be true binders (as proved to be true). Complete data corresponding to the MassTitr read counts per gate, for each sequence at each concentration, are provided have been deposited at GEO with the accession number GSE166938.

A key providing information on which barcode/index pairs correspond to which sample and cell counts for each replicate are provided in *Supplementary file 7*.

## MassTitr Sequence Analysis

The T7-pep library contains many point mutations and frameshifts, in addition to full-length human 36-mers. Consequently, for many hits we identified multiple closely related but non-identical sequences (*Supplementary file 1*). To analyze trends in our dataset, we collapsed hits that we judged to be variants of the same sequence into one representative sequence, leading to a total of 108

non-redundant sequences. If one of the sequence variants matched exactly with a sequence from the human proteome (UniprotKB/Swiss-Prot release 2020_01), that sequence was chosen as the representative sequence (*UniProt Consortium, 2019*). Otherwise, the sequence with the most combined total cell counts was chosen for analysis. To compare these sequences to sequences from the pre-enriched input library, we clustered amino-acid sequences from the input library reads using CD-HIT (*Huang et al., 2010*) with a sequence identity cut-off of 0.7, which effectively collapsed proteomic sequence variants into one representative sequence.

## Identification of Putative Biological Interaction Partners and GO Analysis

To identify putative biological interaction partners among our MassTitr hits we first removed peptides that mapped to a human protein but contained an unnatural FP4 or CXC motif due to frameshift or point mutations (highlighted in yellow in *Supplementary file 1*). Then we identified those peptides predicted to be intrinsically disordered, using an IUPred2A (*Mészáros et al., 2018*) cutoff of >0.4. Finally, we assessed the cytoplasmic localization of hits using cellular component terms from QuickGO (*Binns et al., 2009*). Two proteins from our list (NHSL1 and KIAA1522) did not have any associated terms with our search criteria. For these proteins, we manually curated subcellular localizations from the literature (*Brooks et al., 2010*) and the Human Protein Atlas (*Thul et al., 2017*). For proteins reported to be membrane-bound, such as MIA3 (*Reynolds et al., 2019*), we confirmed that the regions we pulled out as hits were cytoplasmic as annotated in Uniprot (Uniprot Consortium, 2019). GO term enrichments were performed using PANTHER with a Fisher's Exact Test (*Mi et al., 2019*). GO biological process terms with an FDR < 0.05 were designated as enriched.

## Isothermal Titration Calorimetry (ITC)

ITC experiments were performed with two replicates using a VP-ITC microcalorimeter (MicroCal LLC). To prepare samples for ITC, 2.5 mL of 100 µM ENAH EVH1 domain and 1 mL of 800 µM-1.2 mM of SUMO-peptide fusions were dialyzed against 2 L of ITC Buffer (20 mM HEPES pH 7.6, 150 mM NaCl, 1 mM TCEP) at 4 °C overnight. The concentrations of proteins were remeasured after dialysis on the day of the experiment. SUMO-peptide was titrated into the ENAH EVH1 domain at 25 °C. Data analysis and curve fitting were performed with the Origin 7.0 software (OriginLab). The error reported is the fitting error.

## Crystallography

Crystals of ENAH fused at the C-terminus to ABI1 were grown in hanging drops over a reservoir containing 0.1 M sodium acetate pH 4.5 and 2.90 M NaCl. 1 µL of ENAH-ABI1 (250 µM in 20 mM HEPES, 150 mM NaCl, 1 mM DTT) was mixed with 1 µL of reservoir solution, and 3D crystals appeared within two weeks at 18 °C. Diffraction data were collected at the Advanced Photon Source at Argonne National Laboratory, NE-CAT beamline 24-IDE. The ENAH-ABI1 dataset was integrated and scaled to 1.88 Å with AIMLESS and the structure was solved with molecular replacement using ENAH EVH1 structure 5NC7 as a search model. The structure was refined using iterative rounds of model rebuilding with PHENIX and COOT (*Liebschner et al., 2019*; *Emsley and Cowtan, 2004*). *Supplementary file 8* reports refinement statistics. The structure is deposited in the PDB with identifier 7LXE. Note that the ABI1 FP$_8$ peptide is numbered 120–129 in accordance with the ENAH-ABI1 fusion protein numbering.

## Computational Rosetta Modeling

### Input structure

An initial structure modeling bivalent binding was derived from two separate, peptide-bound EVH1-domain structures. Structure 5NC7 (*Barone et al., 2020*) includes a short FP4-containing peptide (chain I) bound to the noncanonical site of the ENAH EVH1 domain (chain D), and the ENAH-ABI1 structure reported in this work provided a model of an FP$_8$ peptide bound at the canonical site, which for modeling was truncated to FP$_4$. These two structures were imported into PyMol and superimposed using the structural alignment function. After verifying that the RMSD for the alignment was low ( ≤ 0.5 Å), the ENAH EVH1 domain of the 5NC7 structure was deleted. The two structures were then merged and exported from PyMol. The resulting PDB file contained an EVH1-domain bound to two

independent FP4 peptides, and this structure was relaxed using the Rosetta Fast Relax protocol run under the default Rosetta energy function REF2015 (*Park et al., 2016*).

## Chain bridging

Linker geometries were modeled using RosettaScripts (*Fleishman et al., 2011*). The Rosetta BridgeChains mover, with the standard Rosetta energy function REF2015 with interchain centroid weights (*interchain_cen*), was used to link the noncanonical- and canonical-site-bound peptides. The insertion motif used for the protocol was "αLX", which specified that the mover should generate backbone coordinates for a loop of length α composed of amino acids derived from any part of Ramachandran space. Starting with $\alpha = 20$, we tested shorter lengths of α until BridgeChains could no longer find a solution. The FP4 motif seeds were anchored in their starting positions using distance constraints generated using the CoordinateConstraintGenerator mover in *RosettaScripts* with a strength/deviation parameter of 0.25 arbitrary units. The noncanonical-site peptide required a secondary set of constraints to maintain the peptide in the observed docking position, so each of its residues was additionally constrained to lie within several angstroms of residue 45 of the EVH1 chain. The final structure, with the two motifs connected by a chain of poly-Val, was then relaxed using the Rosetta *Fast Relax* protocol run under the default Rosetta energy function REF2015. Because bivalent binding was possible in two different orientations that each preserve the polarity of both FP4 motifs as found in the 5NC7 and ENAH-ABI1 structures, this process was done twice, once for each direction.

## Plasmids for immunoprecipitation

Mammalian expression plasmids for 3xFLAG-FHOD1 and 3xFLAG-IFT52 were generated by Gateway recombination of destination and entry vectors using LR Clonase II (Thermo, 11791020) per the manufacturer's instructions. Destination vector pEZYflag was a gift from Yu-Zhu Zhang (Addgene plasmid #18700; RRID:Addgene_18700). Entry vectors for FHOD1 (HsCD00516058), IFT52 (HsCD00399318), and TJAP1 (HsCD00861157) were obtained from DNASU. The mammalian expression plasmid for GFP-mouse ENAH was a gift from Frank Gertler, MIT. Note that mouse and human ENAH have identical EVH1 domains.

## Immunoprecipitation and western blotting

Six-cm plates of 60% confluent HEK293T (confirmed mycoplasma negative and validated through STR testing) were transfected with 2 μg of GFP-ENAH plasmid and either 3xFLAG-FHOD1 or 3xFLAG-IFT52 plasmid, and 12 μg of polyethyleneimine (PEI, linear MW 25,000; Polysciences, 23966). Twenty-four hours post-transfection, cells were lysed with ice cold immunoprecipitation (IP) buffer (10% glycerol, 1% NP-40, 50 mM Tris pH 7.5, 200 mM NaCl, 2 mM MgCl$_2$) supplemented with protease and phosphatase inhibitor cocktails (MilliporeSigma, 539134; Boston BioProducts, BP-479). Crude lysates were incubated on ice for 10 min and clarified by centrifugation at 21,000 g for 10 min at 4 °C. Supernatant protein concentration was determined using a Bradford assay, and 2 mg of protein was loaded into each IP reaction, with 5% input set aside. Lysates were pre-cleared in 5 μL of magnetic Protein A/G bead slurry (Thermo, 88802), and incubated overnight at 4 °C with orbital rotation with mouse monoclonal anti-FLAG (ProteinTech Group 66008–3, RRID:AB_2749837; 5 μg antibody per mg protein). 25 μL magnetic Protein A + G bead slurry was added to each immunoprecipitation assay, and incubated for 2 hr at 4 °C with orbital rotation. Beads were washed three times in ice cold IP buffer with 0.1% Triton X-100 and eluted by boiling at 95 °C for 5 min in 1:1 2 X Laemmeli buffer:IP buffer.

Samples were resolved using SDS-PAGE and 8% acrylamide gels and transferred to Immobilon-FL PVDF membranes (Millipore-Sigma, IPFL00005). Membranes were blocked in Intercept Blocking Buffer (LI-COR, 927–70001) for 1 hr rocking at room temperature and were probed overnight at 4 °C with primary antibodies diluted in blocking buffer and 0.2% Tween-20. Blots were probed sequentially with rabbit polyclonal anti-FLAG (ProteinTech Group 20543–1, RRID:AB_11232216; 1:1000) and the following day with mouse monoclonal anti-GFP (ProteinTech Group 66002–1, RRID:AB_11182611; 1:1000). Membranes were probed with near-infrared fluorescent secondary antibodies goat anti-mouse Alexa Fluor 680 and goat anti-rabbit Alexa Fluor 790 (Thermo, A21057 and A11367 1:20,000), diluted in blocking buffer and 0.2% Tween-20 for 1 hr at room temperature, and then scanned on an Odyssey DLx (LICOR).

## Acknowledgements

This project was supported by NIGMS award R01 GM129007 to AEK, NCI award R01 CA196885-01 to GM, and the Koch Institute Support (core) Grant P30-CA14051 from the National Cancer Institute. Part of this work is based upon research conducted at the Northeastern Collaborative Access Team beamlines, which are funded by the National Institute of General Medical Sciences from the National Institutes of Health (P30 GM124165). The Eiger 16 M detector on the 24-ID-E beamline is funded by an NIH-ORIP HEI grant (S10OD021527). This research used resources of the Advanced Photon Source, a U.S. Department of Energy (DOE) Office of Science User Facility operated for the DOE Office of Science by Argonne National Laboratory under contract DE-AC02-06CH11357. TH was partially supported by NIGMS T32 GM007287 and a fellowship from the Koch Institute for Integrative Cancer Research.

We thank the Koch Institute's Robert A Swanson (1969) Biotechnology Center for technical support, specifically the Flow Cytometry Core Facility for assistance with fluorescence-activated cell sorting. We thank the MIT Structural Biology Core for assistance with X-ray crystallography, the MIT Biophysical Instrumentation Facility for instrumentation resources, and the MIT BioMicroCenter for assistance with high-throughput sequencing. We thank D Whitney for his contributions to the MassTitr technology, and we thank members of the Keating lab and F Gertler for their thoughtful input. We also thank J Tadros and F Gertler for reagents and M Li, S Elledge, and Brigham and Women's hospital for the T7-pep library.

## Additional information

### Funding

| Funder | Grant reference number | Author |
| --- | --- | --- |
| National Institute of General Medical Sciences | GM129007 | Amy E Keating |
| National Cancer Institute | CA196885-01 | Ghassan Mouneimne |
| National Cancer Institute | P30-CA14051 | Ghassan Mouneimne |

The funders had no role in study design, data collection and interpretation, or the decision to submit the work for publication.

### Author contributions

Theresa Hwang, Sara S Parker, Conceptualization, Data curation, Formal analysis, Writing – original draft, Writing – review and editing; Samantha M Hill, Conceptualization, Data curation, Formal analysis, Writing – review and editing; Robert A Grant, Data curation, Formal analysis, Supervision, Writing – review and editing; Meucci W Ilunga, Data curation, Formal analysis; Venkatesh Sivaraman, Software; Ghassan Mouneimne, Conceptualization, Resources, Supervision, Writing – review and editing; Amy E Keating, Conceptualization, Funding acquisition, Supervision, Writing – original draft, Writing – review and editing

### Author ORCIDs

Sara S Parker http://orcid.org/0000-0003-3670-6147
Samantha M Hill http://orcid.org/0000-0002-6454-7430
Robert A Grant http://orcid.org/0000-0002-5072-2867
Ghassan Mouneimne http://orcid.org/0000-0001-8103-4701
Amy E Keating http://orcid.org/0000-0003-4074-8980

### Decision letter and Author response

Decision letter https://doi.org/10.7554/eLife.70680.sa1
Author response https://doi.org/10.7554/eLife.70680.sa2

## Additional files

### Supplementary files

• Supplementary file 1. MassTitr hits and annotated ENAH interactors.

• Supplementary file 2. Dissociation constants for MassTitr peptides derived from human proteins binding to monomeric ENAH EVH1 domain.

• Supplementary file 3. Dissociation constants for ABI1-derived peptides binding to ENAH EVH1 domain.

• Supplementary file 4. Dissociation constants for peptides derived from human proteins binding to monomeric EVH1 domains from ENAH, VASP, and EVL.

• Supplementary file 5. Comparison of affinities of single- and dual-motif peptides for monomeric ENAH WT vs. ENAH R47A.

• Supplementary file 6. Protein constructs for *Native proline-rich motifs exploit sequence context to target actin-remodeling Ena/VASP protein ENAH.*

• Supplementary file 7. MassTitr sample preparation and key.

• Supplementary file 8. Refinement table for ENAH-ABI1 structure.

• Transparent reporting form

• Source data 1. Raw data for *Supplementary file 2*.

• Source data 2. Raw data for *Supplementary file 3*.

• Source data 3. Raw data for *Supplementary file 4*.

### Data availability

Sequencing data have been deposited in GEO under accession code GSE166938. Link to data: https://www.ncbi.nlm.nih.gov/geo/query/acc.cgi?acc=GSE166938. Diffraction data have been deposited in the PDB under accession code 7LXE. All other data generated or analyzed during this study are included in the manuscript and supporting files.

The following dataset was generated:

| Author(s) | Year | Dataset title | Dataset URL | Database and Identifier |
|---|---|---|---|---|
| Keating AE, Hwang T | 2021 | Next Generation Sequencing of bacterial surface-displayed ENAH EVH1 ligand peptides after a FACS titration sort (MassTitr) | https://www.ncbi.nlm.nih.gov/geo/query/acc.cgi?acc=GSE166938 | NCBI Gene Expression Omnibus, GSE166938 |
| Keating AE, Grant RA, Hwang TH | 2021 | ENAH EVH1 domain bound to peptide from ABI1 | https://www.rcsb.org/structure/7LXE | RCSB Protein Data Bank, 7LXE |

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
