## [Editor Report]

The manuscript uses a new screen called MassTitr to display long (36-mer) peptides derived from human proteome to screen for peptides that can bind the EVH1 domain of ENAH protein. About 100 peptides were identified and further analysis identified sequence features that contribute to the binding of EVH1 domain, including an additional proline after the FP4 motif and double FP4 motif. This paper will be of broad interest in the field of proteomics and to scientists interested in how biological interactions achieve specificity.

---

## [Decision Letter]

**Decision letter after peer review:**

Thank you for submitting your article "Native proline-rich motifs exploit sequence context to target actin-remodeling Ena/VASP proteins" for consideration by *eLife*. Your article has been reviewed by 3 peer reviewers, including Hening Lin as the Reviewing Editor and Reviewer #1, and the evaluation has been overseen by Volker Dötsch as the Senior Editor. The following individual involved in review of your submission has agreed to reveal their identity: Linda Nicholson (Reviewer #2).

Essential revisions:

1) Biochemically characterize one or two identified proteins that are not previously known binders and show they indeed bind ENAH in cells.

2) Kd values should be measured in triplicate and When comparing two Kd valudes, statistical analysis should be done.

3) Revise the text according to comments from all reviewers.

Please also note that the three reviewers all felt that two co-submitted manuscripts could be merged into one and make a much stronger manuscript.

*Reviewer #1 (Recommendations for the authors):*

As a stand-alone manuscript, for the manuscript to be suitable for *eLife*, the three weaknesses mentioned in the public review will need to be addressed. I think the authors should test a few binders (at protein level) not previously known and show that they indeed bind to ENAH in cells. This would significantly enhance the manuscript. Mutagenesis studies should be carried out to confirm the conclusions obtained from the structural/sequence analysis. For weakness #3 described in the public review, the authors should clearly state the new insights that are obtained from the MassTitr screening. I am not an expert on screening libraries and thus the authors need to make is clear to non-expert like me. Addressing weakness #1 and #2 could potentially help to address #3 too.

It also appears that the work described in the co-submitted manuscript could help to address these concerns. Thus, another possibility to merge the two manuscript into one.

*Reviewer #2 (Recommendations for the authors):*

This is an important manuscript that applies a new FACS-based screening method to identify binding partners of the EVAH EVH1 domain. A combination of methods were employed to confirm novel hits and investigate interaction mechanisms.

*Reviewer #3 (Recommendations for the authors):*

The impact of this work could be enhanced if key elements from both manuscript were combined into a single, more comprehensive paper. The second manuscript is relatively short but closely related to the first thematically in that it describes MassTitr-identified interaction features.

When comparing KD values that are relatively close (e.g., less than 5-fold difference), it is important to perform those measurements at least in triplicate and then calculate p values (or other comparisons test parameter) to demonstrate that the differences are significant with the experimental parameters. The Supplemental figures indicate BLI assays were performed in duplicate with errors reported as "propagated", but it is not clear what is meant by this. Three independent KD determinations should be made and plotted with error bars that reflect the spread of those values. Comparison among KD values that are larger (e.g., >10-fold difference) are less concerning as they are unlikely to fall within the experimental error of the binding measurement. Notably, the Methods section does not describe the BLI assay.

Please provide more information on how the binning was performed in the MassTitr selection in the Results.

---

## [Author Response]

Essential revisions:1) Biochemically characterize one or two identified proteins that are not previously known binders and show they indeed bind ENAH in cells.

We have carried out co-immunoprecipitation studies validating interactions between ENAH and full-length proteins FHOD1, IFT52, and TJAP1 and added these as Figure 1F and Figure 1—figure supplement 3. We are excited about these results because they provide new hints about ENAH biology. For example, IFT52 and TJAP1 are localized to the cilia and Golgi body, respectively, where ENAH function is not well characterized. FHOD1 is a formin that is thought to support actin filament elongation. Formins and Ena/VASP proteins are known to cooperate in filopodia formation and elongation, but only a few molecular interactions have been characterized. We discuss implications for these results in p. 8-9 of the text.

2) Kd values should be measured in triplicate and When comparing two Kd values, statistical analysis should be done.

All quantitative comparisons are now supported by triplicate measurements. Figure 3, Table 1, Supplementary files 2-5, and their associated source data files have been updated to reflect these new values. We have also carried out a two-sided t test when claiming a difference between two K_D_ values throughout the manuscript. The calculated p-values support our claims that even fold changes small in magnitude are statistically significant and not within error. These results can be found on p. 5, p. 6, and Table 1. Fold change errors in Figure 3B, C were propagated as we divided the average K_D_ of one measurement by the average K_D_ of a separate measurement. We have included a description of the statistical tests used and formula used for error propagation on p.12. Finally, we include both the standard deviation and 95% confidence intervals for all calculated K_D_ values in our source data files.

3) Revise the text according to comments from all reviewers.

We describe changes that we have made to the text based on reviewer suggestions.